# Recombinant Arginine Deiminase from *Levilactobacillus brevis* Inhibits the Growth of Stomach Cancer Cells, Possibly by Activating the Intrinsic Apoptosis Pathway

**DOI:** 10.3390/ijms25084163

**Published:** 2024-04-09

**Authors:** Remilyn M. Mendoza, Ji Hoon Song, Yong Tae Jung, Hyun-Dong Paik, Young-Seo Park, Dae-Kyung Kang

**Affiliations:** 1Department of Animal Biotechnology, Dankook University, Cheonan 31116, Republic of Korea; rmmendoza02@dankook.ac.kr (R.M.M.); sr9412@naver.com (J.H.S.); 2Department of Microbiology, Dankook University, Cheonan 31116, Republic of Korea; yjung@dankook.ac.kr; 3Department of Food Science and Biotechnology of Animal Resource, Konkuk University, Seoul 05029, Republic of Korea; hdpaik@konkuk.ac.kr; 4Department of Food Science and Biotechnology, Gachon University, Seongnam 13120, Republic of Korea; ypark@gachon.ac.kr

**Keywords:** lactic acid bacteria, anticancer, arginine deiminase, apoptosis, genome analysis

## Abstract

The anticancer potential of *Levilactobacillus brevis* KU15176 against the stomach cancer cell line AGS has been reported previously. In this study, we aimed to analyze the genome of *L. brevis* KU15176 and identify key genes that may have potential anticancer properties. Among potential anticancer molecules, the role of arginine deiminase (ADI) in conferring an antiproliferative functionality was confirmed. In vitro assay against AGS cell line confirmed that recombinant ADI from *L. brevis* KU15176 (ADI_br, 5 µg/mL), overexpressed in *E. coli* BL21 (DE3), exerted an inhibitory effect on AGS cell growth, resulting in a 65.32% reduction in cell viability. Moreover, the expression of apoptosis-related genes, such as *bax*, *bad*, *caspase*-7, and *caspase*-3, as well as the activity of caspase-9 in ADI_br-treated AGS cells, was higher than those in untreated (culture medium-only) cells. The cell-scattering behavior of ADI_br-treated cells showed characteristics of apoptosis. Flow cytometry analyses of AGS cells treated with ADI_br for 24 and 28 h revealed apoptotic rates of 11.87 and 24.09, respectively, indicating the progression of apoptosis in AGS cells after ADI_br treatment. This study highlights the potential of ADI_br as an effective enzyme for anticancer applications.

## 1. Introduction

Cancer, characterized by the uncontrolled proliferation of abnormal cells with the potential to invade adjacent tissues and metastasize to other organs, ranks as the second leading cause of death worldwide [1]. Current cancer management strategies, such as surgery, irradiation, and chemotherapy, commonly cause undesirable side effects [1,2]. Therefore, there is a continuous pursuit for safer and more effective alternatives to these conventional cancer treatments [3]. Amino acid deprivation therapy (AADT) is one of the most promising alternative cancer treatments [4]. The metabolic alterations in cancer cells result in an increased demand for energy to sustain their uncontrolled growth, leading to a deficiency in endogenous amino acid supply [5,6]. Consequently, cancer cells become reliant on exogenous sources of amino acids and eventually develop auxotrophy for certain amino acids [6]. This dependency on exogenous amino acids represents a vulnerability in certain cancer types and forms the foundation of AADT [4,7].

Arginine is a semi-essential amino acid that is predominantly obtained from exogenous sources such as diet but can also be synthesized from citrulline via the enzymatic action of arginosuccinate synthase and arginosuccinate lyase in the urea cycle [6]. Arginine is vital for cell proliferation, cell survival, and protein synthesis [8]. In 1947, an increased requirement for arginine in cancer cells was reported [5]. The metabolic machinery in certain human tumor cells is downregulated for arginine synthesis, rendering them auxotrophic for arginine [6,9]. Arginine deiminase (ADI) is responsible for arginine degradation, converting arginine into citrulline and ammonia [5,10]. When cancer cells become auxotrophic for arginine, the activity of ADI is amplified, depleting free arginine available to tumor cells, and potentially leading to cell death [5,6,11]. Several ADIs derived from various microorganisms have been explored for their potential anticancer activities [9,10,12,13].

Various lactic acid bacteria (LAB) and their products, such as exopolysaccharides, bacteriocins, S-layer proteins, peptidoglycan, and ADI, have demonstrated anticancer activity [2,3]. Moreover, different mechanisms of anticancer activity have been reported, including colonization, immunomodulation and anti-inflammation, necrosis, apoptosis, antiproliferation, and angiogenesis inhibition [2]. Nevertheless, information on the anticancer potential of ADIs derived from LAB is limited.

Heat-killed *L. brevis* KU15176 has been reported to exhibit anticancer activity against the stomach cancer cell line AGS by inducing the intrinsic apoptosis pathway [14]. In this study, we sequenced and surveyed the genome of *L. brevis* KU15176 and identified key genes and metabolites that have potential anticancer activity. Here, we highlight the antiproliferative activity of recombinant ADI from *L. brevis* KU15176 against stomach cancer cells. Furthermore, this study provides additional information on the anticancer activity of LAB-derived ADIs.

## 2. Results

### 2.1. Genome Analysis, Identity Verification, and Identification of Potential Anticancer Genes and Gene Products in L. brevis KU15176

The genome assembly of *L. brevis* KU15176 yielded a complete genome of 2,457,403 bp, comprising a single chromosome and three putative plasmids (Figure 1A). *L. brevis* KU15176 has a GC content of 45.88% and harbors 2490 genes, i.e., 2380 coding sequences, 63 transfer RNA genes, 31 miscellaneous RNA genes, 15 ribosomal RNA genes, and one transfer-messenger RNA gene. The calculated average nucleotide identity of the genome of *L. brevis* KU15176 showed that it is >95% similar to that of other *L. brevis* strains available in the NCBI database (e.g., LMT1-73, NPS-QW-145, UCCLBBS449, TMW1.2113, D6, KB290, SRCM101174, SRCM101106) with the highest % identity with *L. brevis* 100D8 with 99% identity, confirming the identity of *L. brevis* KU15176 (Figure 1B). Through mining and manual curation of the genes, we identified ADI, exopolysaccharide, S-layer protein, and peptidoglycan as potential anticancer molecules in the genome of *L. brevis* KU15176 (Table 1).

### 2.2. Arginine Deiminase of L. brevis KU15176

As deduced from the *acrA* gene, the amino acid sequence and structure of the ADI from *L. brevis* KU15176 (ADI_br) were compared with those from *Mycoplasma arginini* (ADI_Ma) and *Lactococcus lactis* subsp. *lactis* ATCC 7962 (ADI_Ll), and *Enterococcus faecium* GR7 (ADI_Ef) (Figure 2), which have been previously characterized to have anticancer activities [9,12,15]. BLASTP alignment revealed that ADI_br exhibited 34.71%, 59.66%, and 63.99% similarity with ADI_Ma, ADI_Ll, and ADI_Ef, respectively. Although ADI_br shares binding and catalytic sites [10] with the three reference ADIs, the predicted structure of ADI_br is more similar to that of ADI_Ll and ADI_Ef which is probably because *Levilactobacillus*, *Lactococcus*, and *Enterococcus* all belong to the order Lactobacillales. The purified recombinant ADI_br had a molecular weight of approximately 46 kDa (Figure 2), which is similar to the reported molecular weight of ADI_Ef.

### 2.3. ADI_br Exhibited Antiproliferative Activity on AGS Cells

A 3-(4,5-dimethylthiazol-2-yl)-2,5-diphenyltetrazolium bromide (MTT) assay using ADI_br against MRC-5 and AGS cells revealed a significant reduction in the viability of AGS cells after treatment with 1–5 µg/mL of ADI_br, while the growth of MRC-5 cells remained unaffected (Figure 3).

### 2.4. ADI_br Potentially Induced Apoptosis in AGS Cells

To determine whether the cytotoxic effect of ADI_br on AGS cells was related to apoptosis, the activity and expression of apoptosis-related genes were analyzed after treatment with ADI_br. Figure 4 shows that ADI_br induces apoptosis in AGS cells. Compared to that in untreated AGS cells, the expression of the *bad* gene in AGS cells was significantly upregulated upon treatment with ADI_br, while the *bax*, *caspase*-3, and *caspase*-7 genes were also slightly upregulated. Additionally, the activity of caspase-9 was slightly higher, whereas caspase-8 activity was lower than that of the control.

### 2.5. Flow Cytometry Confirmed the Induction and Progression of Apoptosis in ADI_br-Treated AGS Cells

To confirm the apoptotic effect of ADI_br, flow cytometry was performed on Annexin V-FITC/PI-stained AGS cells treated with ADI_br. AGS cells treated with ADI_br exhibited increased apoptosis at 24 and 48 h after treatment (Figure 5). The apoptotic rate of ADI_br-treated AGS cells was 11.87 after 24 h, which increased to 24.09 after 48 h of treatment. Untreated AGS cells had an apoptotic rate of 0.85 and 6.41 after 24 and 48 h, respectively. These findings confirmed the induction of apoptosis in AGS cells upon treatment with ADI_br, and further support the potential anticancer activity of ADI from *L. brevis* KU15176.

## 3. Discussion

Heat-killed *L. brevis* KU15176 has previously been shown to induce the intrinsic apoptosis pathway in AGS cells [14]. In this study, we assessed the genome of *L. brevis* KU15176, identifying key genes and metabolites that have potential anticancer activity. Furthermore, we investigated the effects of recombinant ADI_br on AGS cells.

Whole genome sequencing and bioinformatic analyses have been instrumental in the identification of functional genes in LAB [16]. By meticulously curating genes with known anticancer potential from the existing literature, we identified key genes in the genome of *L. brevis* KU15176 that may have potential anticancer activity. These genes include those encoding ADI, S-layer protein, exopolysaccharide, and peptidoglycan [2,3] (Table 1). Sequence alignment revealed that ADI_br shares similar catalytic and binding sites with other ADIs such as ADI_Ma, ADI_Ll, and ADI_Ef, whereas tertiary structure prediction revealed similarities between ADI_br and ADI_Ll and ADI_Ef [10]. *L. brevis*, *L. lactis* subsp. *lactis*, and *E. faecium* are all members of the order Lactobacillales which may explain the closer similarities in the predicted ADI structures from these species. Notably, ADI_Ma has undergone phase II clinical trials as an anticancer agent against hepatocellular carcinoma, metastatic melanoma, mesothelioma, and acute myeloid leukemia [5,17], ADI_Ll has demonstrated cell cycle arrest in another stomach cancer cell line, SNU-1 [15], while *E. faecium* GR7 has demonstrated an antiproliferative effect against the human hepatocellular carcinoma cell line, HepG2 [9].

Our findings indicate that treatment with ADI_br (1–5 µg/mL) reduced AGS cell viability without adversely affecting the proliferation of normal cells (MRC-5). In another study, the purified ADI_Ll (IC50 = 2 μg/mL) and the recombinant ADI_Ll (IC50 = 0.6 μg/mL) also exerted an antiproliferative effect on SNU-1 [15]. The half-maximal inhibitory concentration of ADI_br of the current study, however, has not been tested nor has the activity of the native ADI_br of *L. brevis* KU15176 assayed.

Meanwhile, treatment with ADI_br resulted in a significant increase in the expression of the *bad* gene and a slight upregulation of *bax*, *caspase*-3, and *caspase*-7. Additionally, we observed higher caspase-9 activity in ADI_br-treated cells than in untreated cells. These results suggest that ADI_br induces cell death in AGS cells via the intrinsic apoptosis pathway [1,18]. A simplified description of the apoptosis pathway has been previously provided [1]. The intrinsic apoptosis pathway is initiated by the B-cell lymphoma 2 (Bcl-2) proteins, which are classified into three groups: (1) anti-apoptotic proteins (BCL2, BCL2L1, and BCL2L2), (2) pro-apoptotic proteins (BAD, BIM, BID, and BECLIN1), and (3) pro-apoptotic proteins (BAX and BAK) regulated by the former groups. The activation of group 3 genes leads to mitochondrial outer membrane permeabilization, which consequently leads to the release of cytochrome c (somatic), causing apoptosome formation. In the apoptosome, caspase-9 is activated, followed by the activation of caspase-3, which ultimately leads to apoptosis [1]. The preliminary results of this study suggest that ADI_br is a potential anticancer molecule from *L. brevis* KU15176.

To further substantiate our findings, we monitored Annexin V-FITC/PI-stained AGS cells subjected to ADI_br treatment. Annexin V stains the phosphatidylserine that becomes exposed in the external leaflet of the cell membranes of cells undergoing apoptosis. Staining with Annexin V-FITC and PI enables the segregation and observation of cell populations that were alive (unstained, Q4), undergoing early apoptosis (Annexin V-FITC-stained, Q3), late apoptosis (Annexin V-FITC, PI-stained, Q2), and necrotic cells (PI-stained, Q1) via flow cytometry analysis [19]. In the present study, we observed an increase in the apoptotic rate of AGS cells between 24 and 48 h after ADI_br treatment. Flow cytometric analyses of the stained AGS cells treated with ADI_br confirmed the apoptotic effect of ADI_br (Figure 5).

Microbial ADIs, including those from LAB, have been studied for their anticancer activities against several cancer cell types [9,10,12,13,15,20]. Bacterial ADIs have demonstrated different mechanisms such as cell cycle arrest [15], apoptosis [9,15], and necrosis [11] in several cancer cell lines. ADI is an enzyme typically used in AADT and has been extensively studied as an alternative cancer treatment [5,6,11]. PEGylation is a technique used to enhance ADI stability, as ADI has a short half-life [5]. This instability in ADI was also observed in the present study, as repeated MTT assays using the same dose of the same purified recombinant ADI_br resulted in a lower reduction in cell viability. Furthermore, initial gene expression analysis using a fresher batch of ADI_br led to a significant increase in the expression of apoptosis protease activating factor-1 (*Apaf*-1), *caspase*-3, and *caspase*-7 in ADI_br (2.5 µg/mL)-treated AGS cells Appendix A. In line with this, we increased the concentration of ADI_br used in subsequent experiments (i.e., flow cytometry and caspase-8 and caspase-9 activity assay) to 10 µg/mL to compensate for the possible degradation and instability of ADI.

The present study highlights the ease of identifying and narrowing down candidate genes or molecules for wet lab experimentation to verify specific functions when genomic data are accessible and previous research on such biological functions is available. Notably, this study shows that ADI from *L. brevis* KU15176 exerted an antiproliferative activity against stomach cancer cells. However, whether ADI played a role and contributed to the previously reported anticancer activity of heat-killed *L. brevis* KU15176 warrants further investigation. As seen in Table 1, several anticancer-associated genes/metabolites have been identified in the genome of *L. brevis* KU15176 and the role of these in the exhibited anticancer activity of heat-killed *L. brevis* KU15176 is yet to be determined. In addition, whether the reported anticancer activity of *L. brevis* KU15176 was due to a single metabolite, or a concerted activity of genes/metabolites also seeks further investigation.

Nonetheless, this study provides evidence that recombinant ADI from *L. brevis* KU15176 can induce apoptosis in AGS likely through the activation of the intrinsic apoptosis pathway. It also provides additional information on bacterial ADIs that have potential anticancer activity. While this study has several limitations including: the enzymatic activity of ADI_br under varying physicochemical conditions, such as pH and temperature, has not been optimized; and the half-maximal inhibitory concentration of ADI_br has not been tested; this study provides additional information on the antiproliferative activity of LAB-derived ADIs against stomach cancer cells and offers a promising LAB-derived enzyme that may be further explored as an anticancer agent.

## 4. Materials and Methods

### 4.1. Cultivation of Bacterial and Mammalian Cells

*L. brevis* KU15176, previously isolated and characterized [14,21], was cultured in d Man–Rogosa–Sharpe (MRS) broth (BD Biosciences, Franklin Lakes, NJ, USA) at 37 °C for 24 h. The stomach cancer cell line AGS (KCLB21739, Seoul, Republic of Korea) was cultured in Roswell Park Memorial Institute (RPMI) 1640 medium (Gibco^TM^, New York, NY, USA), whereas MRC-5 cells (KCLB10171, Seoul, Republic of Korea) were cultivated in Dulbecco’s modified Eagle’s medium (DMEM, Gibco^TM^). Both culture media were supplemented with 10% fetal bovine serum (HyClone^TM^, Logan, UT, USA), and 1% Zellshield^®^ (Minerva Biolabs, Berlin, Germany). Mammalian cell lines were cultivated at 37 °C in a humidified atmosphere containing 5% CO_2_.

### 4.2. DNA Extraction, Whole Genome Sequencing, Assembly, Annotation, and Gene Mining

The genomic DNA of *L. brevis* KU15176 was extracted using the conventional phenol-chloroform protocol and subjected to whole genome sequencing at CJ Biosciences Inc. (Seoul, Republic of Korea) using the PacBio Sequel System (Pacific Biosciences, Menlo Park, CA, USA) NGS method. The raw sequencing reads of *L. brevis* KU15176 were processed with Canu v.2.2 [22] and assembled using Flye v.2.9 [23]. The identity of *L. brevis* KU15176 was confirmed using the Python module pyani v.0.2.12 [24] with the ANIb option against available genomes in the NCBI database. Genome annotation was performed using RAST (https://rast.nmpdr.org, accessed on 26 February 2022 [25,26]) and Prokka v1.14.6 [27]. Assembly statistics were generated using QUAST v.5.2.0 [28], and a genome map was generated using DNAPlotter (Artemis v.18.2.0) [29]. Genes associated with anticancer activity in LAB were manually curated based on the existing literature. The tertiary structure of ADI_br was predicted using the Phyre2 v.2.0 web portal [30], and multiple sequence alignments of the ADI_br (*arcA*) with reference ADI sequences were established using the Clustal Omega web server (https://www.ebi.ac.uk/jdispatcher/msa/clustalo, accessed on 28 February 2024) [31].

### 4.3. Cloning of ADI_br

Forward (5′-AA**CTGCAGGCTAGC**ATGACAAGTCCGATTCAC-3′) and reverse (5′-GC**TCTAGACTCGAG**AAGGTCTTCTCGAACTAATG-3′) primers targeting the ADI in the genome of *L. brevis* KU15176 were designed for amplification and cloning of ADI_br_ in *Escherichia coli* BL21 (DE3), and were synthesized by Bionics Co., Ltd (Seoul, Republic of Korea). The restriction enzyme sites *PstI-NheI* and *XbaI-XhoI* are indicated in bold font. PCR amplification was performed using Ex Taq DNA polymerase (TaKaRa, Maebashi, Japan) in a PCR Thermal Cycler Dice^®^ Gradient (TaKaRa, Maebashi Japan). The thermocycling conditions were as follows: initial denaturation at 95 °C for 5 min, followed by 35 cycles of denaturation at 95 °C for 30 s, annealing at 58 °C for 30 s, and extension at 72 °C for 1 min. The reaction concluded with a final extension step at 72 °C for 7 min. The amplified *arcA* gene was cloned into the pet21b+ plasmid and transformed into *E. coli* BL21 (DE3) cells. Plasmid DNA was extracted from the clones using a Plasmid Mini-Prep Kit (Dyne Bio Co., Ltd., Seongnam, Republic of Korea), and the extracted plasmids were sequenced at Bionics Co., Ltd. (Seoul, Republic of Korea). Plasmids from clones containing the correct ADI_br sequence were extracted using the QIAGEN^®^ Plasmid Midi Kit (Venlo, The Netherlands).

### 4.4. Overexpression of ADI_br

The pet21b+ plasmid containing the correct ADI_br sequence was transformed into competent *E. coli* BL21 (DE3) cells. Successful transformants were cultured in 100 mL Luria-Bertani broth supplemented with 100 µg/mL ampicillin and incubated at 37 °C with shaking at 200 rpm. Cell growth was monitored by measuring the absorbance at 600 nm using a spectrophotometer (Shimadzu, Kyoto, Japan) until the optical density reached 0.5. ADI_br expression was induced by adding 1 M of Isopropyl β-D-1-thiogalactopyranoside and incubating for 3.5 h at 30 °C. Subsequently, cells were harvested using centrifugation (4000× *g* for 15 min at 4 °C). The cells were washed twice and resuspended in lysis buffer (50 mM NaH_2_PO_4_, 300 mM NaCl, pH 7), and disrupted using an ultrasonic oscillator (VCX750, Sonics, WA, USA) at 30% amplitude. Subsequently, cellular debris were harvested using centrifugation at 14,000× *g* for 30 min at 4 °C. Denaturing lysis buffer (lysis buffer with 5 M urea and 0.1 mM phenylmethylsulfonyl fluoride, PMSF) was used to dissolve the pellet for 1 h on ice. The urea-soluble fraction was collected using centrifugation at 14,000× *g* for 20 min.

### 4.5. Purification of ADI_br

ADI_br in the urea-soluble fraction was purified following the protocol for Ni-NTA Agarose (QIAGEN, MD, USA). Briefly, 1 mL Ni-NTA matrix (bed volume, 0.5 mL) was equilibrated with equilibration buffer (lysis buffer with 10 mM imidazole). Approximately 5 mL of the urea-soluble fraction was added to the equilibrated Ni-NTA matrix, and the mixture was incubated on ice with shaking at 200 rpm for 1 h. Subsequently, the protein-Ni-NTA mixture was loaded onto a column and the flow-through was collected. The protein was washed with 5-bed volumes of wash buffer (lysis buffer containing 20 mM imidazole). Finally, ADI_br was eluted using an elution buffer (lysis buffer containing 20 mM imidazole, 20% glycerol, 0.1 mM PMSF, and 1 mM dithiothreitol [DTT]). To remove the imidazole, the eluted ADI_br underwent buffer exchange using the Amicon^®^ Ultra-15 centrifugal filter unit (Sigma Aldrich, St. Louis, MO, USA). Finally, purified ADI_br was suspended in lysis buffer (pH 7) containing 20% glycerol and 1 mM DTT. The fractions were separated using sodium dodecyl sulfate-polyacrylamide gel electrophoresis.

### 4.6. MTT Assay

The effect of ADI_br treatment on the proliferation of AGS and MRC-5 cells was evaluated using the MTT assay [14]. Briefly, AGS and MRC-5 cells were seeded in separate 96-well plates at a density of 2 × 10^4^ cells/well. The plates were incubated overnight at 37 °C with 5% CO_2_. Subsequently, the culture medium was removed and ADI_br (1–5 µg/mL) in the respective cell culture media was added to each well. Cells incubated with only DMEM and RPMI were used as controls. The plates were incubated for 44 h. After incubation, the culture media was aspirated, and 100 µL of 0.5 mg/mL MTT reagent (Invitrogen, Waltham, MA, USA) was added to each well. The plates were incubated again for 4 h. Residual MTT reagent was carefully aspirated, and 200 µL of dimethyl sulfoxide was used to dissolve the formazan salts. Wells with DMSO only were used as blanks. The absorbance at 570 nm was measured using the SpectraMax M2e (Molecular Devices, San Jose, CA, USA), and the cytotoxicity of ADI_br on AGS and MRC-5 cells was calculated using the following formula:$$Cytotoxicity\left(\%\right)=\left(1-\frac{A_{sample}}{A_{control}}\right)\times 100$$

### 4.7. RNA Extraction and RT-qPCR

The expression of apoptosis-related genes was determined using RT-qPCR, with the β2-microglobulin gene (*B2M*) serving as the internal control [32]. Total RNA was isolated from ADI_br (5 µg/mL)-treated AGS cells using the RNeasy^®^ Mini Kit (QIAGEN, Hilden, Germany). Subsequently, cDNA was synthesized using the AccuPower^®^ RT Premix (BIONEER, Daejeon, Republic of Korea) following the manufacturer’s protocol. The expression of specific genes, including *bad*, *bax*, *caspase*-3, and *caspase*-7, was determined using SYBR Green PCR Master Mix, and qPCR was performed on an IQ5 Multicolor Real-Time PCR Detection System (Bio-Rad, Hercules, CA, USA) following the delta-delta Cq method for analysis. Table 2 lists the primers used for RT-qPCR.

### 4.8. Caspase-8 and Caspase-9 Colorimetric Assay

The activity of caspase-8 and caspase-9 in AGS cells treated with 10 µg/mL ADI_br for 48 h was determined using the Elabscience^®^ (Houston, TX, USA) Caspase 8 and Caspase 9 Activity Assay Kit following the manufacturer’s instructions. Caspase activity was expressed as a percentage increase in enzyme activity.

### 4.9. Apoptosis Analysis Using Flow Cytometry

Apoptosis in AGS cells treated with ADI_br (10 µg/mL) for 48 h was monitored using flow cytometry. The Dead Cell Apoptosis Kit (Thermo Fisher Scientific, Waltham, MA, USA) was used for the assay. Briefly, AGS cells treated with ADI_br were harvested, washed with cold Phosphate-Buffered Saline (PBS), resuspended in 100 µL of 1 × annexin binding buffer to a concentration of ~1 × 10^6^ cells/mL, and stained with Annexin V-FITC and Propidium Iodide for 15 min. Subsequently, 500 µL of 1 × annexin binding buffer was added to the stained cells. In all, 10,000 cells were analyzed, and the FL-1 and FL-2 channels in a BD Accuri^TM^ C6 Plus Flow Cytometer (BD Biosciences, NJ, USA) were used to observe the stained cells. FlowJo^TM^ (version 10.9.0) was used to analyze and visualize flow cytometry data.

### 4.10. Statistical Analysis

Statistical analysis was conducted on the means of triplicate measurements using the analysis of variance implemented in GraphPad Prism 8.4.2.

## 5. Conclusions

In this study, we compared ADI from *L. brevis* KU15176 with that from *M. arginini, L. lactis* subsp. *lactis*, and *E. faecium* GR7 which have exhibited anticancer activity. Our findings revealed that purified recombinant ADI from *L. brevis* KU15176 exhibited antiproliferative activity against AGS cells. Our results also suggest that ADI may induce the intrinsic apoptosis pathway in AGS cells. Notably, we observed an upregulation in the expression of intrinsic apoptosis pathway-related genes, such as *bad*, *bax*, *caspase*-3, and *caspase*-7, and increased caspase-9 activity. Flow cytometric analyses of ADI_br-treated AGS cells indicated apoptosis induction, as evidenced by the cell-scattering behavior of Annexin V-FITC/PI-stained ADI_br-treated cells. This study highlights the potential of ADI_br as an effective enzyme for anticancer applications. However, it is imperative to improve enzyme stability to mitigate issues associated with inconsistent results that may arise during long-term storage. Therefore, further research may provide valuable insights and advancements in using ADI_br in cancer therapy.

## Figures and Tables

**Figure 1 ijms-25-04163-f001:**
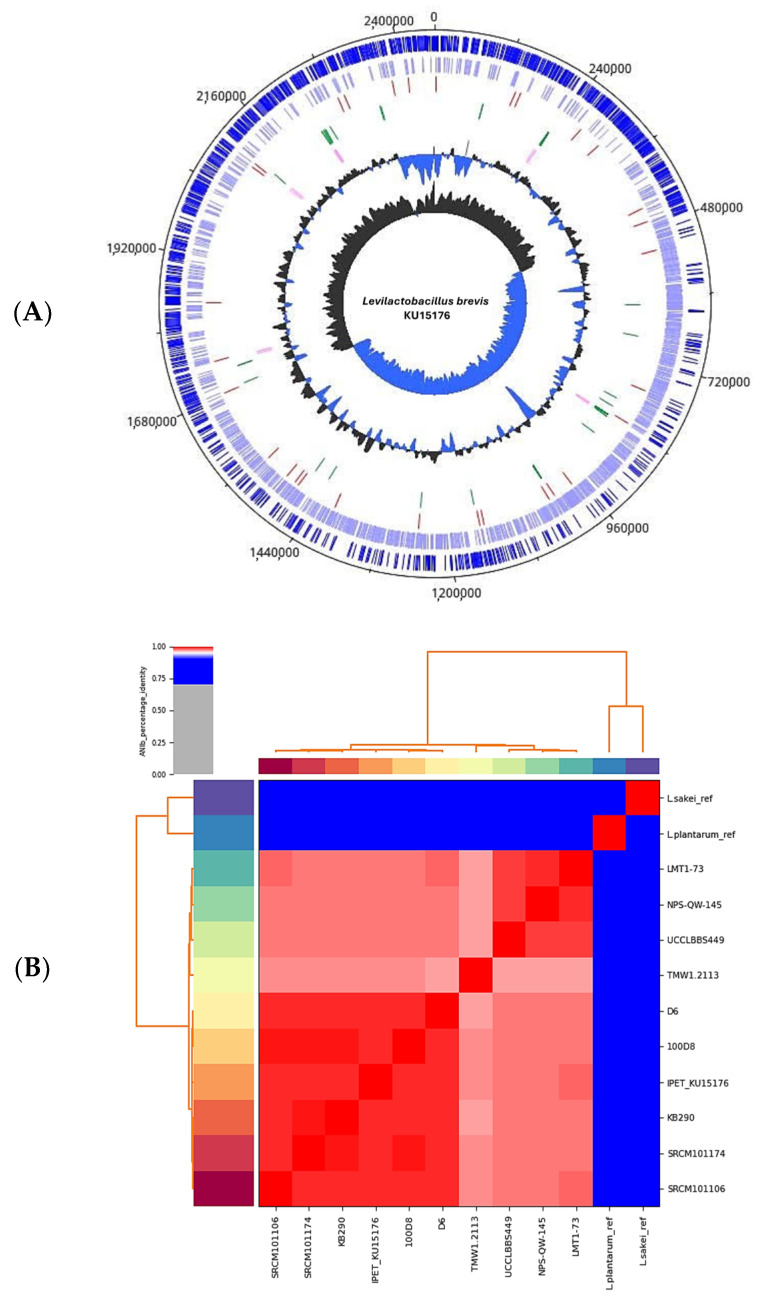
**Genome analysis of *L. brevis* KU15176.** (**A**) The genome map and features of *L. brevis* KU15176. From the outside going in, it includes CDS in the forward strand, CDS in the reverse strand, miscellaneous RNA, transfer RNA, ribosomal RNA, transfer-messenger RNA, GC plot, and GC skew. The genome map was generated using DNAPlotter implemented in Artemis (v.18.2.0) while the features were provided by QUAST (v.5.2.0) and Prokka (v1.14.6). (**B**) Percentage identity heatmap of *L. brevis* KU15176 against reference *L. brevis* strains (LMT1-73, NPS-QW-145, UCCLBBS449, TMW1.2113, D6, 100D8, KB290, SRCM101174, SRCM101106) and *L. plantarum* SK151 (L.plantarum-ref) and *L. sakei* CBA3614 (L.sakei-ref) strains as outliers verified the identity of *L. brevis* KU15176 that is >95% similar to that of the reference *L. brevis* strains. The heatmap was generated by implementing the ANIb method in pyani (v0.2.12).

**Figure 2 ijms-25-04163-f002:**
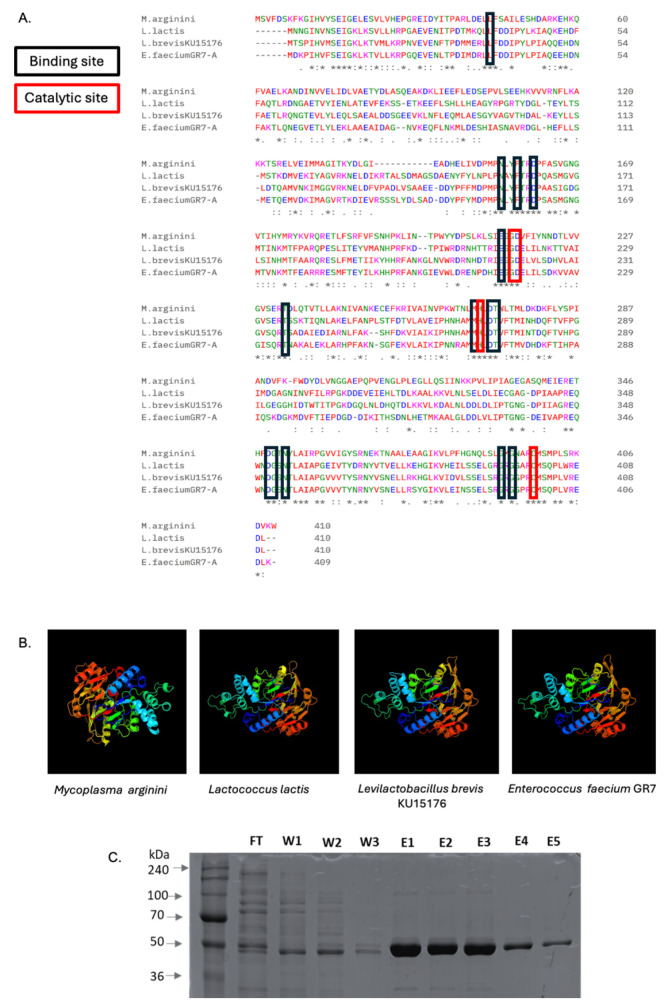
**Arginine deiminase.** (**A**) Multiple sequence alignment (MSA) of ADI from *M. arginini*, *L. lactis* subsp. *lactis* ATCC 7962, *L. brevis* KU15176, and *E. faecium* GR7. The binding and catalytic sites are boxed in black and red, respectively. MSA was generated using the Clustal Omega webserver. Meanwhile, (**B**) the predicted structures of the ADIs are also shown which are generated using the Phyre2 webserver. (**C**) SDS-PAGE of purified recombinant ADI_br confirmed its molecular weight at about 46kDa. Lane 1: Molecular weight marker (DokDo-Mark^TM^, Daejeon, Republic of Korea), FT: Flow-through, W1-W3: washings, E1-E5: eluted ADI_br.

**Figure 3 ijms-25-04163-f003:**
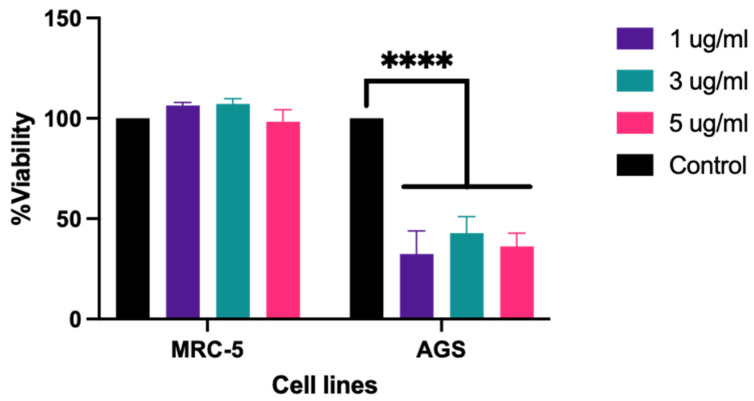
**ADI_br exhibited an antiproliferative effect on AGS cells**. Treatment of ADI_br reduced the viability of AGS cells (**** *p* < 0.0001) while having no significant effect on the viability of MRC-5 cells. Dulbecco’s modified Eagle’s medium (DMEM) and Roswell Park Memorial Institute (RPMI) were used as negative control in the setups with MRC-5 and AGS, respectively, while wells with dimethyl sulfoxide (DMSO) only were used as blanks. Data presented are means of three replicate readings. Statistical analysis was conducted using two-way ANOVA implemented in GraphPad Prism 8.4.2.

**Figure 4 ijms-25-04163-f004:**
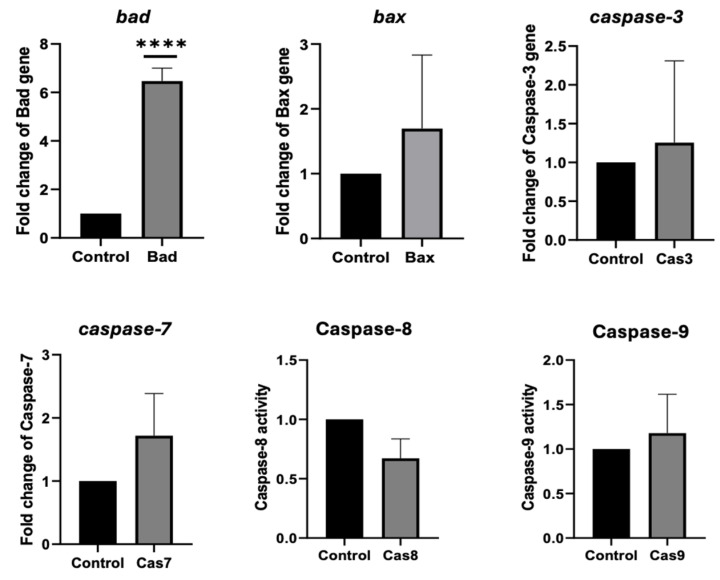
**ADI_br potentially induced intrinsic apoptotic machinery in AGS cells**. Treatment of ADI_br upregulated the expressions (*bad* [**** *p*<0.0001], *bax*, *caspase*-3, *caspase*-7) and increased the activity (caspase-9) of the intrinsic apoptosis-related genes. β2-microglobulin (*B2M*) was used as the internal control for the analysis following the *delta-delta* Cq method. Data presented are means of three replicate readings. Statistical analysis was conducted using one-way ANOVA implemented in GraphPad Prism 8.4.2.

**Figure 5 ijms-25-04163-f005:**
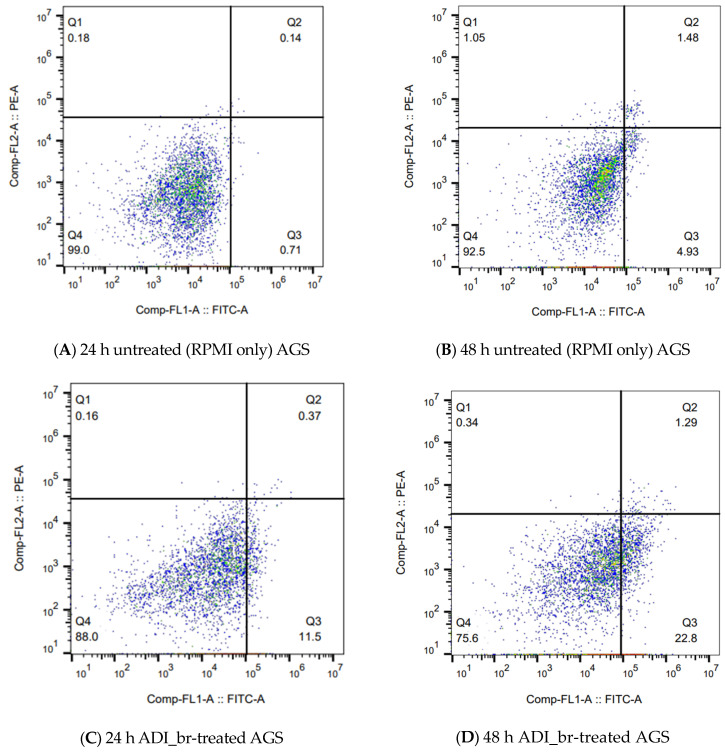
**Flow cytometry analyses confirmed the progression of apoptosis in ADI_br-treated AGS cells.** The behavior of untreated (**A**,**B**) and ADI_br-treated AGS cells (**C**,**D**) stained with Annexin V-FITC and PI was observed by flow cytometry. The cell population represented in quadrants are as follows: Q1: dead/necrotic cells, Q2: late apoptotic cells, Q3: Early apoptotic cells, and Q4: live cells. The red/orange dots correspond to areas of higher cell densities, yellow is mid-range, and green/blue indicates areas with lower cell densities. FlowJo^TM^ was used to analyze and generate images of cell-scattering behavior.

**Table 1 ijms-25-04163-t001:** Putative contributory anticancer gene and gene products in *L. brevis* KU15176.

Anticancer-Associated Metabolites/Genes	Annotation/Gene Product
**Arginine deiminase**	
*arcA*	Arginine deiminase
**S-layer protein**	
LbrevisKU15176_01469	S-layer protein
**Exopolysaccharide (Biosynthesis)**	
*YwqE*	Mn-dependent protein-tyrosine phosphatase
*EpsC*	Tyrosine-protein kinase transmembrane modulator
*YwqD*	Tyrosine-protein kinase
*EpsE*	Undecaprenyl-phosphate galactosephosphorotransferase
*EpsF*	Putative glycosyltransferase
*EpsH*	Putative glycosyltransferase
*EpsL*	Putative sugar transferase
LbrevisKU15176_01006	dTDP-glucose 4,6-dehydratase
LbrevisKU15176_01013	UDP-galactopyranose mutase
*MurJ*	Flippase
*YwqC*	Putative capsular polysaccharide biosynthesis protein
**Peptidoglycan (Recycling and Biosynthesis)**	
*glmM*	Phosphoglucosamine mutase
*glmU*	N-acetylglucosamine-1-phosphate uridyltransferase
*murA1_1*	UDP-N-acetylglucosamine 1-carboxyvinyltransferase 1
*murA1_2*	UDP-N-acetylglucosamine 1-carboxyvinyltransferase 1
LbrevisKU15176_00604	N-acetylmuramoyl-L-alanine amidase
LbrevisKU15176_00626	N-acetylmuramoyl-L-alanine amidase

**Table 2 ijms-25-04163-t002:** Primers used for the measurement of the expression of apoptosis-related genes.

Genes	Sequence (5′-3′)	Product Length (bp)	Annealing Temperature (°C)
** *B2M* **			
Forward	AGATGAGTATGCCTGCCGTG	105	55.5
Reverse	GCGGCATCTTCAAACCTCCA		
** *bax* **			
Forward	CCCCCGAGAGGTCTTTTTCC	160	54
Reverse	TGTCCAGCCCATGATGGTTC		
** *bad* **			
Forward	CCTTTAAGAAGGGACTTCCTCGCC	325	54
Reverse	ACTTCCGATGGGACCAAGCCTTCC		
***caspase*-3**			
Forward	AGCGGATGGGTGCTATTGTG	172	54
Reverse	ACACCCACCGAAAACCAGAG		
***caspase*-7**			
Forward	ACGATGGCAGATGATCAGGG	85	54
Reverse	GGTCTGGCTTAGCATCCACT		

*B2M*: Beta-2 microglobulin, *Bax*: BCl-2 associated X; Bcl-2 agonist of cell death.

## Data Availability

The whole genome sequence of *L. brevis* KU15176 is deposited in the GenBank database under accession nos. CP143802-CP143805 with BioSample accession no. SAMN39613023, and BioProject project accession no. SAMN39613023.

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
