# Peer review of "Recombinant Arginine Deiminase from Levilactobacillus brevis Inhibits the Growth of Stomach Cancer Cells, Possibly by Activating the Intrinsic Apoptosis Pathway"

_ijms, 2024, doi:10.3390/ijms25084163_

Round 1

Reviewer 1 Report

Comments and Suggestions for Authors

This manuscript reports on the genome analysis of a single L. brevis strain, in which the arginine deiminase enzyme was considered as a candidate molecule to be tested for inhibiting some cancer-related activities in an in vitro assay using AGS cells. The gene was amplified and cloned in E. coli, and the product was overexpressed and used against AGS cells. At a concentration of 5 microg/mL the purified His-tagged enzyme was shown to inhibit the viability of AGS cells by 65%. Additionally, some apoptosis-related genes and enzymes were shown to be expressed in higher amounts than in controls, as did some enzyme activities.

General comments:

1.- The manuscript reports on in vitro assays showing the arginine deiminase enzyme from L. brevis KU15176 has inhibitory activity on the expression of enzymes involved in cancer. Due to the multifaceted sides of cancer, the expression “anticancer activity”, “anticancer potential”, and other related expressions should be utilized with care throughout the manuscript. In this manuscript, what is reported is an anti-proliferative activity of an enzyme. My suggestion is to soften expressions such as those in lines 16, 24, 62, 206, 214, 332, and might be others. In addition, the name of the strain in the title is not necessary, unless the enzyme from this specific L. brevis strain is different from those in other strains; a Blast analysis will suffice to check this homology.

2.- In the L. brevis strain, neither the activity of the native enzyme nor the expression of the gene has been demonstrated. Further, the activity of the recombinant enzyme on arginine was not determined. The activity of the protein has been inferred based solely on the homology against enzymes from unrelated species (M. arginine and L. lactis). Do you think this is enough to be sure you are working with the arginine deiminase from L. brevis? Is the activity of the enzyme in L. brevis KU15176 strong enough to influence arginine metabolism in the gut?

3.- There are a lot of references within the Results section (lines 74-75; 97-99, etc.). These references are not required here, as in most cases you are reporting your results. Avoid discussions in this section and leave them for the real Discussion.

4.- The study of the caspases involves both gene expression (caspase 3 and 7) and enzyme activities (caspase 8 and 9). Why two different approaches were utilized?

5.- Most of the figures are not very informative and/or lack appropriate explanations that help understand the real meaning. These should be reformed or deleted.

6.- The list of references includes some which are incomplete, such as numbers 8 and 11; in addition, several layouts were noted in the titles (see 14, 16, 29, 33, for instance); names of the journals are either in full or abbreviated (such as in 14 and 15); title of the journal in capitals (such as in 30), etc. Please, revise and harmonize the references following the Journal Instructions to Authors guidelines.

7.- Lines 181-184; the hypothesis that ADI_br may be one of the effectors of the anti-proliferative effect of L. brevis KU15176, I think, comes from previous studies (literature) with the same enzyme from other microorganisms (M. argini, L. lactis, and E. faecalis). What you have seen before is the antiproliferative activity of the strain against cancer cells.

Specific comments:

1.- In the title, genome analysis (DNA) and arginine deiminase (a protein) do not match. Genome analysis identified genes.

3.- Line 17;…confirmed in an in vitro assay against the AGS cell line.

3.- Line17; please, include overexpression in E. coli

4.- Table 1; replace “Possible anticancer metabolites” with “Anticancer-associated metabolites/genes”.

5.- Figure 1B is not required, as these results have been already reported in paragraph 68-75 (please, mind the space in line 74 between the word and the square bracket).

6.- Figure 1C is completely useless. Further, some details are lacking, for instance, the code of the reference strains, or not explained, such as the red-to-blue color guide below the chromatograms (there is some green and yellow color, which are not in the central square).

7.- Line 89; As deduced from the acrA gene, the amino acid sequence and structure of the ADI from L. brevis were compared…

8.- Lines 96-99; apparently, the ADI enzyme from E. faecalis is very similar to that of L. brevis. Why the sequence of the enzyme from this species (and perhaps others) was not included in Figure 2?

9.- Figure 2B; as shown in the pictures, the structure of the ADI enzyme from L. lactis and L. brevis are highly similar. This should be stressed in the previous description of the enzymes.

10.- Figure 3; why color bars were not used in this figure? The greyscale is difficult to read.

11.- Line 116; please, mind antiproliferative.

12.- Line 140; add at in from of the 24 and 48 h.

13.- Line 144; mind the non-italicized L. brevis.

14.- Figure 5; unless a proper explanation is given in the text, the figure is useless. Which is the part of the population that predicts cell death and why?

15.- Line 167; is the concentration of the ADI recombinant enzyme (5 microg/Ml) of physiological relevance? Is this concentration attained in any biological fluid? Please, discuss.

16.- Line 180; mind the space between the word and the square bracket.

17.- Lines 185-191; this paragraph either repeats the results obtained or reports results that should be transferred to that section.

18.- Line 206; this study suggests that ADI might play a…

19.- Line 210; has not been tested/assayed.

20.- Line 214; anticancer activity has neither been tested nor demonstrated.

21.- Line 237; ADI tertiary structure was predicted…

22.- Line 242; ADI gene was called ascA in Table 1; maintain the naming.

23.- Line 243; primers were designed for amplification and cloning.

24.- Line 244; was the cloning done directly in E. coli BL21(DE3)? Please, state it.

25.- Line 250; the amplified arcA gene.

26.- Lines 253-255; why were the plasmids containing the right gene sequence overexpressed? I do not understand this step.

27.- Lines 266-267; are you sure the cells were harvested by centrifugation after the treatment with ultrasounds? A majority of cells should be disrupted, and cellular debris should be centrifuged and removed.

28.- Lines 289-290; spell in full the abbreviations DMEM and RPMI (if not stated before).

29.- Lines 300-301; the gene encoding the beta2-microglobulin (B2M) serving as the internal… The same observation can be made for the footnote of Table 4; please, make a distinction between genes and proteins.

30.- Table 2; genes had been written without a capital initial in the text. For homogenization, please, follow the same writing all along the manuscript. In this regard, in Figure 4, the names of the genes should also follow this layout (but not those of the Caspase 8 and 9 that are OK).

Comments on the Quality of English Language

English usage is quite OK; the text just needs to be proofread for typing mistakes and grammatical errors.

Author Response

We are grateful for the constructive criticisms and suggestions given for the improvement of our manuscript. Attached is our point-by-point response to each comment in the hopes that these improved and answered the unclear details of our manuscript.

Again, thank you very much.

Reviewer 2 Report

Comments and Suggestions for Authors

Review of the manuscript entitled Genome analysis identified arginine deiminase from Levilactobacillus brevis KU15176 that inhibits the growth of stomach cancer cells, possibly by activating the intrinsic apoptosis pathway.

The authors confirmed the anticancer potential of Levilactobacillus brevis KU15176 against the stomach cancer cell line, AGS, that was reported previously. In this study, they analyzed the genome of L. brevis KU15176 and identified key genes responsible for its anticancer properties. Further they examined the role of arginine deiminase (ADI) with anticancer functionality among potential anticancer molecules. Overexpressed recombinant ADI from L. brevis KU15176 (ADI_br, 5 µg/mL) exerted an inhibitory effect on AGS cell growth, resulting in a 65.32% reduction in cell viability.

I had several problems with this manuscript. Although the manuscript about the effect of arginine deiminase (ADI) on cancerous cell growth, several facts are not clear, and messed up.

I may accept that the enzyme ADI is responsible for the induction of apoptosis of the AGS cells, but the manuscript failed to give answers how a bacterial cytoplasmic enzyme ADI can get out from L. brevis, and how ADI would be endocytosed by the cancerous cells. Without this knowledge I cannot believe that the way of action of the degradation of Arg can occur.

It should be clearly shown how the enzyme ADI is secreted from L. brevis, and specifically enter the AGS cells, but not MRC-5 cells.

In addition, I need an explanation, why ADI_Ma and ADI_L1 do not prefer AGS cells and block the cell proliferation, if the ADIs have the same enzyme activity.

Further, earlier Hwang et al. (2022) published that heat inactivated L brevis blocks the cancerous cell growth. If the authors really think that the heat inactivated cells block the cell growth by ADI enzyme they should prove that this enzyme remains active upon heat inactivation.  Hwang et all. (2022) did not claim such things. I would consider the work of Kuntz et al., (2008: Oligosaccharides from human milk induce growth arrest via G2/M by influencing growth-related cell cycle genes in intestinal epithelial cells).

Sequencing the genome of L. brevis is one of the most important result of the article. However I cannot agree with this sentence #lines 204-206:

"This study highlights the ease of identifying candidate genes or molecules for specific functions when genomic data are accessible and previous research on such biological functions are available."

The knowledge of any genome does not provide any biological functions, just the study of proteins can give, and show the real biological functions.

Please add control to the MTTassay. The MTT assay is a colorimetrical assay for assessing redox potential of the cellular NAD(P)H. Therefore, it is necessary to test the applied buffer, medium, and ADI solution. It cannot be accepted to measure the cell viability without adequate controls.

Author Response

We appreciate the constructive criticisms and suggestions given for the improvement of our manuscript. Attached is our point-by-point response to each comment in the hopes that these improved and answered the unclear details of our manuscript.

Again, thank you very much. 

Round 2

Reviewer 1 Report

Comments and Suggestions for Authors

I appreciate all modifications and changes in the text as a response to my review comments and suggestions, for which I would like to congratulate the authors. The text is almost ready, and I am just pointing out a few minor (editorial) details that, in my opinion, will improve the final version of your article.

1.- The reporting of the ANI value against some other L. brevis strains as >95% is quite OK; but it will also be nice to state the highest ANI value with the reported strains. An even better is to have this value against the type strain of the species. This can be done, for instance, in the Type Strain Genome Server of the DSMZ culture collection (https://tygs.dsmz.de/); it takes less than two minutes.

2.- In line 92, the references to L. plantarum and L. sakei strains are lacking.

3.- In line 98; it is the first appearance of E. faecium and should, therefore, be spelled in full (Enterococcus faecium).

4.- The expression "overproduced" for a plasmid is kind of odd. Strategies have been developed for plasmid amplification (in certain cases and plasmids), but I do not think this is the case either in this work. My suggestion is to delete "overproduced and" in line 278 and "overproduced" in line 280.

Author Response

We are grateful for the dedication and the time allotted in reviewing our manuscript for its improvement. Below is our response for the remaining comments of the reviewer.

Reviewer 2 Report

Comments and Suggestions for Authors

2nd review of the manuscript entitled Recombinant Arginine deiminase from Levilactobacillus brevis inhibits the growth of stomach cancer cells, possibly by activating the intrinsic apoptosis pathway

Previously I asked the authors how the arginine deiminase is released from the host bacterium cells. Is it secreted, or product of degradation?

They answered: Arginine deiminases essentially work extracellularly. As the extracellular arginine levels decrease via the activity of ADI, the amount of free arginine becomes insufficient for the growth of cancer cells that requires higher amounts of amino acid to support their abnormal cell growth. Especially in the cancer cells that auxotrophic to arginine, the insufficient arginine can lead to hampered proliferation and cell death. Normal cells such as MRC-5 do not have elevated arginine requirement hence do not rely to exogenous supply to sustain cell growth.

Unfortunately, this is not an answer for the question.

Further, it was requested to add control to the MTTassay, because the MTT assay is a colorimetrical assay for assessing redox potential of the cellular NAD(P)H. Therefore, it is necessary to test the applied buffer, medium, and ADI solution. It cannot be accepted to measure the cell viability without adequate controls.

The DMSO does not belong to the medium, nor to the ADI buffer. Please repeat it, and measure the MTT assay on the buffers, media, as well.

My suggestion the authors should separate into a different article the effect of arginine deiminase on the cancerous cells, and into a bioinformatics article, where they present the genomial sequence of Levilactobacillus brevis. The two topics are pretty altered from each other. Arginine deiminase is a common enzyme, it can also  be found in e.g. E. coli.

Comments on the Quality of English Language

It is OK.

Author Response

We are grateful for the dedication and the time allotted to reviewing our manuscript for its improvement. Below is our response to the remaining comments of the reviewer.

Round 3

Reviewer 2 Report

Comments and Suggestions for Authors

The manusript was not changed significantly.

Comments on the Quality of English Language

The English is correct.

Author Response

To Reviewer 2

We appreciate the time dedicated to the review of our manuscript. We are grateful for the constructive criticisms, insights, comments, and suggestions given for the improvement of our manuscript. Please see the attached document summarizing the modifications we have made to address the comments.
